# Selection and Validation of Reference Genes for RT-qPCR Analysis in *Aegilops tauschii* (Coss.) under Different Abiotic Stresses

**DOI:** 10.3390/ijms222011017

**Published:** 2021-10-13

**Authors:** Adeel Abbas, Haiyan Yu, Xiangju Li, Hailan Cui, Jingchao Chen, Ping Huang

**Affiliations:** 1Key Laboratory of Weed Biology and Management, Institute of Plant Protection, Chinese Academy of Agricultural Sciences, Beijing 100193, China; adeel.abbas92@yahoo.com (A.A.); yuhaiyan2103@163.com (H.Y.); hlcui@ippcaas.cn (H.C.); jchcheng@ippcaas.cn (J.C.); 2Institute of Environment and Ecology, School of Environment and Safety Engineering, Jiangsu University, Zhenjiang 212013, China; huangjiehp@ujs.edu.cn

**Keywords:** *Aegilops tauschii* Coss, gene expression analysis, abiotic stresses, gene validation, reverse-transcribed quantitative PCR, gene responsive to stress

## Abstract

*Aegilops tauschii* (Coss.) is an aggressive and serious annual grass weed in China. Its DD genome is a rich source of genetic material and performs better under different abiotic stress conditions (salinity, drought, temperature, etc.). Reverse-transcribed quantitative polymerase chain reaction (RT-qPCR) is a reliable technique for reference gene selection and validation. This work aimed to evaluate the stability of reference gene expression in *Ae. tauschii* under different abiotic stresses (salinity, drought, hot, and cold) and developmental stages (seedling and development). The results show that the ubiquitin-conjugating enzyme E2 36-like (*UBC36*) and protein microrchidia 2-like (*HSP*) are the most stable genes under control and salinity conditions, respectively. Under drought stress conditions, *UBC36* is more stable as compared with others. Glyceraldehyde-3-phosphate dehydrogenase (*GADPH*) is the most stable reference gene during heat stress conditions and thioredoxin-like protein (*YLS*) under cold stress condition. Phosphate2A serine/threonine-protein phosphatase 2A (*PP2A*) and eukaryotic translation initiation factor 3 (*ETIF3*) are the most stable genes at seedling and developmental stages. Intracellular transport protein (*CAC*) is recommended as the most stable gene under different abiotic stresses and at developmental stages. Furthermore, the relative expression levels of *NHX1* and *DREB* under different levels of salinity and drought stress conditions varied with the most (*HSP* and *UBC36*) and least (*YLS* and *ACT*) stable genes. This study provides reliable reference genes for understanding the tolerance mechanisms in *Ae. tauschii* under different abiotic stress conditions.

## 1. Introduction

*Aegilops tauschii* Coss. (2*n* = 2x = 14) is a wild species and reported as a donor D genome in cultivate wheat (*Triticum aestivum* L.=AABBDD) [1,2]. It has the widest geographic distribution in many wheat-growing countries from Turkey to west Afghanistan and Central Asia, and it has adapted to different diversified conditions, performing better against heat, drought, and salinity stresses [3]. This weed spreads in China’s main wheat production areas and competes for light, water, and resources [4]. D genome of *Ae. tauschii* is a rich source of genetic material, and many traits can be used for wheat improvement programs, including drought, heat, and salinity [5,6]. Researchers are trying to evaluate these traits and have identified the genes for wheat crop improvement. Gene expression analysis is important in crop improvement programs and helps understand plants’ molecular mechanisms against different abiotic stresses [7].

Reverse-transcribed quantitative PCR (RT-qPCR) is deemed a highly suitable strategy for gene expression studies and can detect target genes’ expression. The RT-qPCR method is the most important technique and a common way to examine genes’ relative expression patterns in many crops and weeds [8]. Different challenges in gene expression studies through RT-qPCR relate to reference gene expression stability and to RNA and complementary deoxyribonucleic acid (*cDNA*) quality [9]. The selection and the use of an unstable reference gene create problems and unpredictable changes in gene expression studies in RT-qPCR [10]. Different kinds of reference genes, such as glyceraldehyde 3-phosphate dehydrogenase (*GADPH*), actin (*ACT*), elongation factor-1 alpha (*EF1α*), ubiquitin-conjugating enzyme E2 36-like (*UBC36*), and tubulin (*TUB*), have been used in many experiments but have not been uniformly expressed in the critical conditions [11]. An ideal and appropriate reference gene presents stability in different plant developmental stages and under different biotic and abiotic stress conditions [12]. Many studies reported in the past demonstrated that there is not a universal reference gene for all plant species and all experiments [13]. *ACT* has been reported as the most stable in tomato plants but proved least stable in cucumber under salinity stress, while *18S rRNA* has been reported as a stable reference gene in rice, though unstable in some other crops [14,15]. Similarly, many other reference genes have been tested in various species. In general, it is necessary to select appropriate reference genes for different species, different treatments, and different experimental conditions [16].

NormFinder, RefFinder, BestKeeper, and geNorm have been developed and used for reference genes evaluation [17]. Because of their different algorithms, geNorm, NormFinder, and BestKeeper produce different results, while RefFinder provides a comprehensive ranking of reference genes [18]. Reference genes for many crops, including wheat, maize, soybean, poplar, eucalyptus, and peanut, have been reported, but the number is small for weeds [19]. Although several molecular studies on abiotic mechanisms in *Ae. tauschii* have been conducted [20,21], validation of reference genes in this weed have been rarely reported. This study evaluated the stability of several reference genes and validated them under different abiotic conditions and developmental stages. In the current study, we evaluated gene stability and its validation under different abiotic stresses and at developmental stages in *Ae. tauschii*. We evaluated 12 reference genes—actin (*ACT*), glyceraldehyde-3-phosphate dehydrogenase (*GAPDH*), thioredoxin-like protein (*YLS*), polypyrimidine tract-binding protein homolog 1-like (*PTB*), protein microrchidia 2-like (*HSP*), S-adenosyl methionine decarboxylase proenzyme (*SAMDC*), ubiquitin-conjugating enzyme E2 36-like (*UBC36*), elongation factor 1 alpha (*EF1**α*), serine/threonine-protein phosphatase 2A (*PP2A*), eukaryotic translation initiation factor 3 (*ET1F3*), intracellular transport protein (*CAC*), and peptidyl-prolyl cis-trans isomerase *CYP19*-4-like (*CYP)*—under different abiotic stresses (salinity, drought, heat, and cold stress) and developmental stages (seedling and development).

## 2. Results

### 2.1. Reference Genes Selection

Twelve reference genes were selected, including *ACT*, *GAPDH*, *YLS*, *PTB*, *HSP*, *SAMDC*, *UBC36*, *EF1-alpha*, *PP2A*, *ET1F3*, *CAC*, and *CYP* (Table 1). All primers showed amplification specificity by gel electrophoresis (Appendix A). Then, the candidate primer pairs were detected by RT-qPCR based on a standard curve and efficiency. Genes’ *R^2^* were stably greater than 0.98, RT-qPCR efficiency (E = −1 + 10^(−1/slope)^) between 104% and 117% (Table 1) and annealing temperature of PCR was 48–57. The results under different abiotic stresses and developmental stages in *Ae. tauschii* was based on the estimation of Cq. The Cq value of each gene was indicated by the cycle at which the PCR amplifications entered the logarithmic phase. The reference genes with lower Cq values were considered to have higher expression than the other genes. The 12 candidate reference genes under different abiotic conditions and developmental stages showed diverse values in their Cq, ranging from 20.11 to 37.59 (Figure 1). Under salinity, the *PP2A* gene showed significantly higher expression, with Cq values ranging from 23.45 to 26.65 (Figure 1). On the other hand, *ACT* showed lower expression, with values ranging from 31.90 to 37.05. Under control conditions, Cq values ranged from 23.72 to 25.05, *ETIF3* showed high expression, while *CYP* showed Cq ranging from 35.43 to 37.59. Similarly, under different abiotic stresses and developmental stages (Figure 1a–g), all reference genes showed different Cq values.

### 2.2. geNorm Analysis

geNorm analysis was used to calculate reference genes’ expression stability (M) under different abiotic stresses and developmental stages. Genes with the lowest M value are considered the most stable genes, while the least stable genes have higher M values. Reference genes with M value < 1.5 can be utilized in further studies (Figure 2). *PP2A* and *CAC* were denoted as the most stable genes with an M value of 0.10, while *CYP* was the least stable gene under control conditions. Under salinity stress conditions, *PP2A* and *HSP* were the most stable genes, with M values of 0.06, while *YLS* was the least stable gene, with an M value of 1.20. *UBC36* was recorded as the most stable gene with an M value of 0.15 under drought stress conditions, while the *ACT* was considered the least stable gene, with an M value of 1.05. Under heat stress conditions, *UBC36* was detected as the most stable gene, with an M value of 0.10, while *HSP* was considered the least stable gene, with a high M value of 1.37. Under cold stress conditions, *GADPH* and *CAC* were recorded as the most stable genes, with the lowest M value, while *ETIF3* was considered the least stable gene, with a high M value. In the seedling stage, *PP2A* and *ETIF3* were considered the most stable genes, with the lowest M value of 0.01, respectively. On the other hand, *ACT* was denoted as the least stable gene, with the highest M value of 0.52. At the developmental stage, *PP2A* and *PTB* were the most stable genes, with M values of 0.69, respectively, while *UBC36* was the least stable gene with an M value of 2.53. Overall, *CAC* and *HSP* proved to be the most stable genes, with an M value of 1.61. The optimal number of reference genes required for accurate normalization was computed. All values were obtained for the pairwise variation between consecutive normalization factors.

Pairwise variation (Vn/Vn + 1) in genes to minimize the implementation of qPCR expression analysis with a cutoff value of < 0.15 (Figure 3). Seven to nine reference genes were valid for normalization under different abiotic stresses in *Ae. tauschii*, while under drought stress treatment and development stages genes showed variations.

### 2.3. NormFinder Analysis

NormFinder measured the stability value (M) of reference genes using evaluations under different abiotic stresses and developmental stages in *Ae. tauschii.* (Figure 4). *UBC36*, *PP2A, UBC36, CAC, YLS, PP2A, ETIF3*, and *PTB* were the most stable reference genes under control, salinity, drought, heat and cold, and under both growing (seedling and developmental) stages.

### 2.4. BestKeeper Analysis

BestKeeper analysis base on standard deviation (SD) and coefficient of variance (CV) was conducted to measure and evaluate the stability of reference genes. The lower SD and CV values are considered more stable. In addition, an SD value of less than 1 is acceptable for reference gene selection. The values with lower SD had higher expression levels and were the most stable, while a higher SD value indicated the least stable reference gene (Table 2). Under different abiotic stresses (control, salinity, drought, heat, cold) and developmental stages, *UBC36, CAC, EFI-alpha, SAMDC, GADPH, UBC36*, *CYP*, and *HSP* had lower SD values and were considered the most stable reference genes, respectively.

### 2.5. RefFinder Analysis

RefFinder is a web tool that is used to calculate the heterogeneity in the results of geNorm, NormFinder, and BestKeeper. A comprehensive ranking showed that *UBC36, HSP, UBC36, UBC36, YLS, PP2A*, and *ETIF3* were the most stable genes, with lower M values in *Ae. tauschii* under different abiotic stresses and different developmental stages (Table 3).

### 2.6. Reference Gene Validation

The expression level of target genes *(NHXI* and *DREB*) was determined under different levels of salinity and drought stress conditions. Both most and least stable genes were chosen under salinity and drought stress conditions. Viewing the superiority in stability, *HSP* was considered further for the expression profiling of *NHXI* which, compared with expression at 50 Mm and 150 NaCl, the expression level appeared 1.6 times and 5.7 times at 50 and 150 mM NaCl, respectively (Figure 5a), while the expression level of the least stable gene, *YLS,* was only 1.4 times and 3.1 times, respectively. Under different drought stress conditions, the expression level of the target gene *DREB,* with most stable reference gene *UBC36* under drought C_1_ = (control, C_2_ = 75 g L^−1^ PEG-6000 and C_3_ = 100 g L^−1^ PEG-6000) conditions was 4.97, 2.73, and 1.66 times and 1.07 times under different water field capacity treatment, respectively (Figure 5b). However, a difference was presented between the results for the most and least stable reference genes. The expression level of *DREB* with the least stable gene, *ACT*, showed 2.16, 1.29, and 0.85 times under different water field capacity treatment.

## 3. Discussion

Abiotic stresses, such assalinity, drought, heat, and cold, significantly affect plant survival, growth, and biomass production [22,23,24]. Plants have developed various mechanisms against various stresses and significantly alter the gene expression profile [25]. In the qRT-PCR analysis, the evaluation and selection of reference genes and PCR conditions are crucial and important prerequisites of gene expression profile analysis [26]. In the present study, we used 12 reference genes (*ACT*, *GAPDH*, *YLS*, *PTB*, *HSP*, *SAMDC*, *UBC36*, *EF1-alpha PP2A, EF1F3, CAC,* and *CYP*) and examined stability and validation under different abiotic stress conditions and developmental stages. The RT-qPCR efficiency of our study was between 104% and 117%; a similar range was used for *ACT* and *GADPH*, with 91% and 118% [27]. These reference genes were also used in different plants for gene expression profiling. Expression stability evaluated by the geNorm, NormFinder, and BestKeeper was not constant, as has been reported in previous studies [28]. In our study, *PP2A* and *HSP* were reported to be the most stable reference genes under salinity stress by geNorm analysis, but with NormFinder and geNorm they performed unsatisfactorily. Therefore, the comprehensive ranking through RefFinder results provides help in selecting appropriate reference genes. Through geNorm, reference genes were determined based on the pairwise variation between sequentially ranked genes (Vn/Vn+1), with the cutoff value of 0.15. When the Vn/n+1 value was below 0.15, no additional genes were required for accurate normalization [29].

*UBC36* and *GADPH* were chosen for normalization and *UBC36* showing high stability in *Arabidopsis thaliana*, on other hand, its expression profile in soybean and rice varied under different developmental stages [30,31]. *HSP* and *PP2A* showed more stability in our study, and *GADPH* was found to be the least stable reference gene under salinity stress conditions. In *Cajanus cajan*, *HSP90* and *UBC* showed stability under salinity stress conditions, and several studies recognized them as stable stress candidate genes specifically used as internal controls [32]. Many studies have focused on investigating the resistance mechanism in different plants under abiotic stress conditions by using RT-qPCR. However, a very small number of studies have reported reference gene selection in weeds under abiotic conditions [33]. *UBC36* appeared to be more stable under drought conditions in *Ae. tauschii*, with low M values according to NormFinder. Similarly, in chickpea plants, *EF1α* and *UBC36* were reported to be the most stable reference genes under drought stress conditions. Our analyses also robustly suggested that *ACT* and *YLS* are the least stable reference genes under drought stress conditions in *Ae. tauschii*. The housekeeping gene *GADPH and UBC36* was reported to be the most stable gene in *Triticum durum* under different abiotic stress conditions (including drought) [34]. Under cold stress conditions, *HEL* was reported to be the most stable reference gene in *Robinia pseudoacacia* [35]. Considering the results in our study under cold stress conditions, *YLS* was found to be the most stable gene in *Ae. tauschii*. *GAPDH* plays a vital role in carbohydrate metabolism and DNA repair [36]. *GADPH* showed more stability under cold stress and had ranked first in the BestKeeper analysis, which showed that our results precisely agreed with prior results.

Similarly, in *Robinia pseudoacacia* and *Camellia sinensis, GADPH* showed the most stability under various abiotic stresses [35,37]. *PP2A* and *ETIF3* played important roles and had the most stable genes under the *Ae. tauschii* seedling and development stages. Likewise, *UBC36* and *EF1α* retained the most stable genes in *Avena sativa* under various abiotic stresses [38]. In total, *CAC* and *HSP* were the most important and showed more stability in *Ae. tauschii* under various abiotic stress conditions. Likewise, in goosegrass, *eIF-4* and *ALS* showed stability under different herbicide stresses [19]. We also observed such apparent effects in our research, although previous studies showed differing results. Considering the importance of the validation of reference genes, we selected the target gene *NHX1* to evaluate the expression level under salinity stress conditions [39]. Comparison between most and least stable genes showed differences in fold change expression in the target genes. With *HSP* as a reference gene, *NHX1* showed significantly higher expression under 50 mM NaCl and 150 mM NaCl. Similarly, *HSP* showed more stability in pigeon pea under salt stress conditions [32]. The expression level of *DREB* was evaluated with the most and least stable genes *UBC36* and *ACT*, respectively, under drought stress conditions. The expression level of *DREB* showed a higher fold change with the most stable reference gene, which showed that selection of a suitable reference gene is essential for the accuracy of the RT-qPCR results. Similarly, validations of reference genes were analyzed by the expression level of the internal control target gene [40]. These results showed that *HSP* and *UBC36* were the most stable reference genes under salinity and drought stress conditions. This study is important and will aid in obtaining accurate results at the gene expression level in *Ae. tauschii*. Similarly, reference gene validation was analyzed by the expression level of the internal control target gene [41]. Our results show that *HSP* is the most stable reference gene under salinity stress conditions. This study will aid in attaining accurate results by using the most appropriate reference genes to find the relative gene expression levels under different kinds of stresses in *Ae. tauschii*. Moreover, our study is limited to the selection of some known genes for the validation of reference genes in *Ae. tauschii*. It needs further study to deeply analyze the reference genes using RNA-Sequencing, which can further verify and can provide a variety of reference genes under abiotic stress and developmental stages in *Ae. tauschii.*

## 4. Materials and Methods

### 4.1. Plant Materials

*Ae. tauschii* genotypes were collected from five different provinces (Shandong, Shanxi, Shaanxi, Hebei, and Henan) of China [41]. Seeds were sown in plastic pots containing sandy clay under greenhouse conditions, and plants were selected based on their performance on abiotic stress. When the seedlings reached the two- to three-leaf stages, different stresses were applied, including salinity stress (50 mM and 150 mM NaCl) and drought stress (75 g L^−1^ PEG-6000 and 100 g L^−1^ PEG-6000) applied in *Ae. tauschii* with four replications [42,43]. For heat and cold stress, plants were exposed in the chamber at −4 °C and 40 °C for five days and then transferred to an environmental growth chamber maintained at 27 ± 1 °C with four replications. For every stress application, a controlled environment was maintained throughout the experiments. Leaf samples were harvested after 10 days of stress application and stored at −80 °C. For developmental stages, the leaves were harvested at the seedling (at the two-leaf stage) and the development stage (at the four- to five-leaf stages) and stored at −80 °C.

### 4.2. RNA Isolation and cDNA Synthesis

RNA was extracted by RNA prep Pure Plant Kit (Cat. #DP432, Tiangen Biotech Beijing CO., LTD, Beijing, China) by using 1 g of leaf samples. Genomic DNA was completely eliminated using RNase-Free DNase I (Tiangen Biotech Beijing CO., LTD, Beijing, China), according to the manufacturer’s instructions. The integrity was checked by 1.0% agarose gel electrophoresis. The purity and concentration of RNA was estimated with a NanoDrop spectrophotometer (NanoDrop Technologies, Wilmington, DE, USA). The cDNA was synthesized by using the Fast Quant RT kit (Cat#KR116-02, Tiangen Biotech Beijing CO., LTD, Beijing, China) with 1 μg of total RNA. All samples were stored at −80 °C.

### 4.3. Reference Genes Selection, Amplification, and Primer Design

Twelve reference genes were tested, including *ACT, GAPDH, YLS, PTB, HSP, SAMDC, UBC36, EF1-alpha, PP2A, ET1F3, CAC*, and *CYP.* Reference gene sequences were obtained from the National Center for Biotechnology (NCBI) database for primer design. Forward and reverse primers (5′-3′) of reference genes were designed by Beacon Designer software and used within the exon region because primers are specific for the amplification of cDNA that contains intron between exon-exon junctions. Primer sequence, amplicon length, melting temp, and RT-qPCR efficiency are presented in Table 1.

### 4.4. RT-qPCR Analysis

An ABI 7500 RT-qPCR machine was used to detect reference genes’ expression level by utilizing SYBR green (Applied Biosystems, Foster City, CA, USA). The total volume of 20 μL of PCR mix containing 10 μL of SYBR Green PCR Master Mix, 1 μL of cDNA, 0.6 μL of each primer, 0.6 μL dye, and 7.2 μL of ddH_2_O was used. The cycling conditions of the RT-qPCR were used to obtain the melt curve temperature: 10 min at 95 °C, 40 cycles of 95 °C for 15s, 57–58 °C for 32 s, with an increase of 0.5 °C every 5 s conditions and extended at 72 °C for 30 s. Three technical and four biological replicates were used.

### 4.5. Data Analysis

BestKeeper, NormFinder, and geNorm were used to evaluate reference gene stability. RefFinder was used as a web tool to calculate the comprehensive ranking of reference genes under different abiotic stresses and at developmental stages. The Cq value of RT-qPCR was used in geNorm, BestKeeper, and NormFinder. GeNorm renewed the Cq values into relative ones by the formula 2^−ΔCt^ (ΔCt = the corresponding Cq value—minimum Cq). GeNorm also calculates the pairwise variation (Vn/Vn+1) with a recommended cut-off value of 0.15.

### 4.6. NHX1 and DREB Expression under Salinity and Drought Stress Condition

For reference gene stability, the expression level of the *NHX1* (Na^+^ transporters involved in plant salt tolerance) and *DREB* (dehydration-responsive element-binding protein 1E) genes was determined under different salinity (50 and 150 mM NaCl) and drought stress (75 g L^−1^ PEG-6000 and 100 g L^−1^ PEG-6000) conditions. Primers for *NHX1* were designed in RT-qPCR with Beacon Designer software. *NHX1* was amplified using the primer pair *NHX1-F*: CCGCAACCAAGTAGAGAAG / *NHX1-R*: GAGCATCATAAGAGCAACCT, while *DREB* was amplified with the pair *DREB-F*: CGAGTCTGTTGATGAGTCT / *DREB-R* 5′ TTCTGTAGTAAGTGCTTGCTA, respectively. The expression levels of *NHX1* and *DREB* were normalized by using the most and least stable genes under salinity and drought stress conditions.5. Conclusions

We determined the reference genes’ stability under abiotic stresses and at developmental stages of *Ae. tauschii*. Results show that *HSP, UBC36, YLS, UBC36, PP2A*, and *ETIF3* were the most stable genes under the salinity, drought, cold, heat, seedling, and development stages of *Ae. tauschii*, respectively. This study will promote future studies in gene expression and will help better understand the responses of *Ae. tauschii* under different abiotic stresses and developmental stages. Results of this study will be helpful in *Ae. tauschii* gene expression studies, which will lead to a wheat crop improvement program.

## Figures and Tables

**Figure 1 ijms-22-11017-f001:**
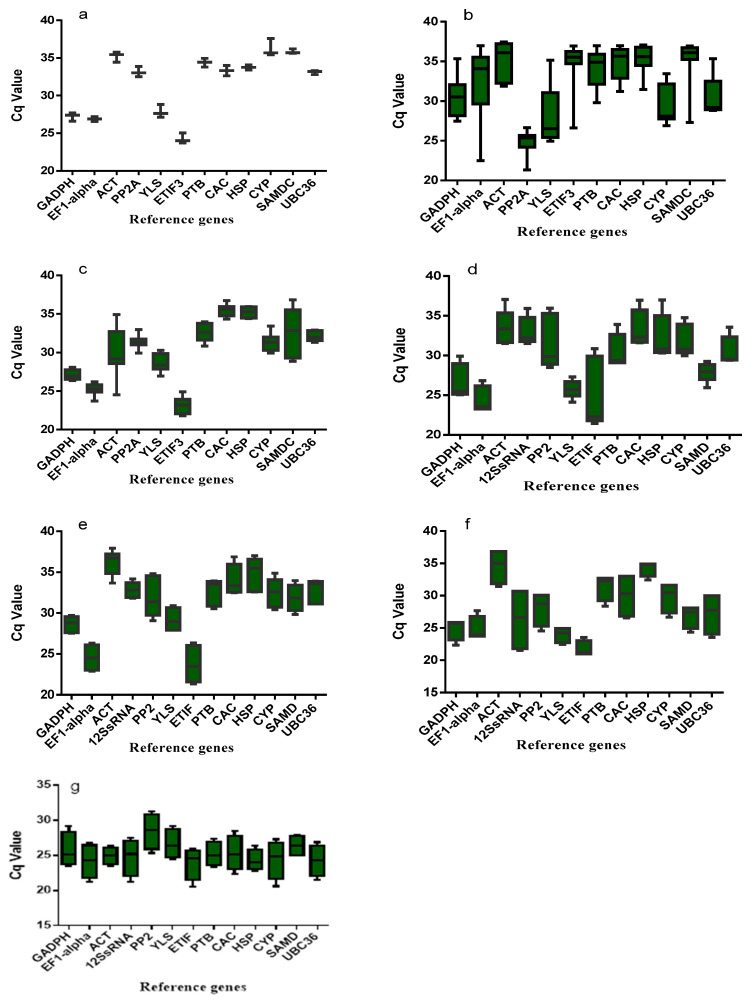
Expression profile of reference genes under control (**a**), salinity (**b**), drought (**c**), heat (**d**), cold (**e**), seedling (**f**), and development stage (**g**). Boxes indicate the 5th/95th percentiles, lines across the boxes depict the medians, squares. ACT = actin, GAPDH = glyceraldehyde-3-phosphate dehydrogenase, EF1-alpha = elongation factor 1 alpha, SAMDC = S-adenosylmethionine decarboxylase proenzyme, UBC36 = ubiquitin-conjugating enzyme E2 36-like, CYP = peptidyl-prolyl cis-trans isomerase CYP19-4-like, PTB = polypyrimidine tract-binding protein homolog 1-like, CAC = intracellular transport protein, PP2A = serine/threonine protein phosphatase 2A, ETIF3 = eukaryotic translation initiation factor3, HSP = protein microrchidia 2-like, YLS = thioredoxin-like protein.

**Figure 2 ijms-22-11017-f002:**
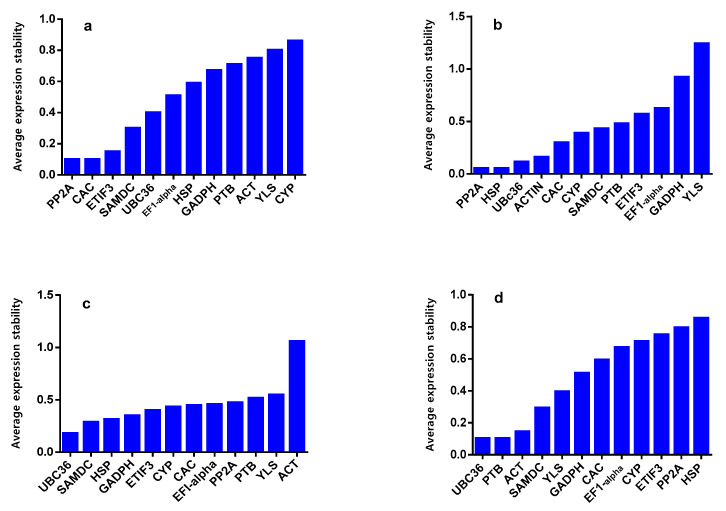
Average expression stability (M) calculated by geNorm under control, abiotic stress conditions, and different developmental stages in *Ae. tauschii*. (**a**) Control, (**b**) salinity, (**c**) drought, (**d**) heat, (**e**) cold, (**f**) seedling (**g**) development, and (**h**) total. *ACT* = actin, *GAPDH* = glyceraldehyde-3-phosphate dehydrogenase, *EF1-alpha* = elongation factor 1 alpha, *SAMDC*= S-adenosylmethionine decarboxylase proenzyme, *UBC36* = ubiquitin-conjugating enzyme E2 36-like, *CYP* = peptidyl-prolyl cis-trans isomerase *CYP19-4*-like, *PTB* = polypyrimidine tract-binding protein homolog 1-like, *CAC* = intracellular transport protein, *PP2A* = serine/threonine protein phosphatase 2A, *ETIF3* = eukaryotic translation initiation factor3, *HSP* = protein microrchidia 2-like, *YLS* = thioredoxin-like protein.

**Figure 3 ijms-22-11017-f003:**
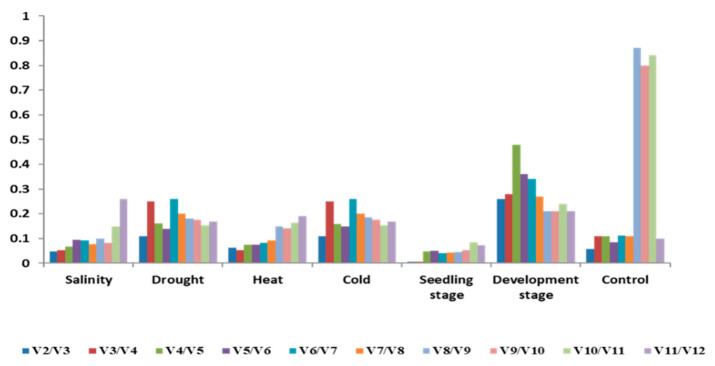
Pairwise variation (V) of reference genes required under control, salinity, drought, cold, heat, seedling stage, and at the developmental stage.

**Figure 4 ijms-22-11017-f004:**
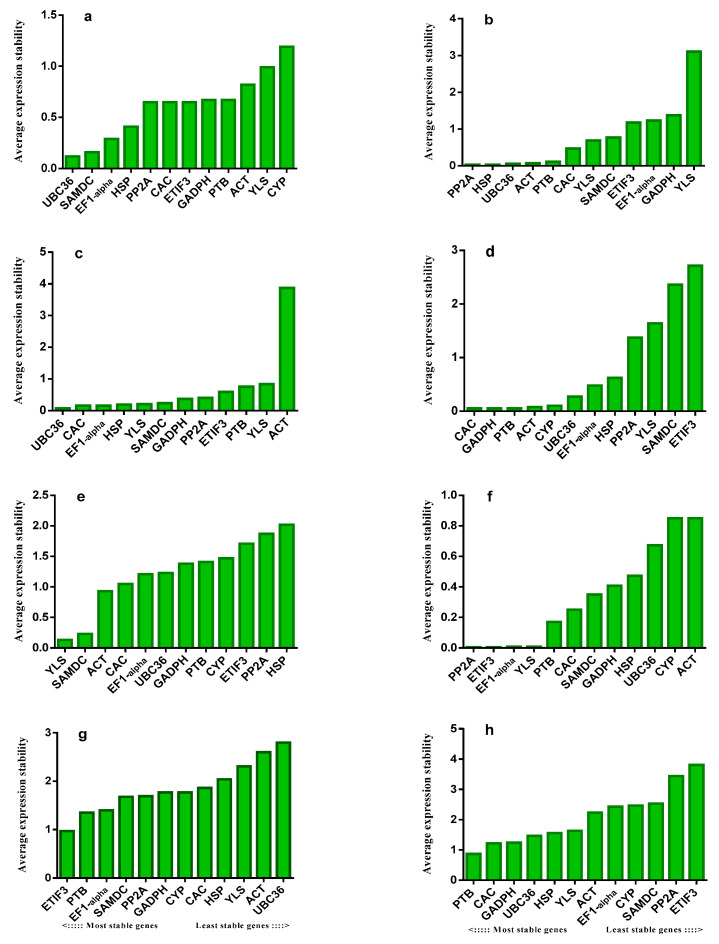
Average expression stability (M) calculated by NormFinder under control, abiotic stress conditions and different developmental stages in *Ae. tauschii.* (**a**) control, (**b**) salinity, (**c**) drought, (**d**) heat, (**e**) cold, (**f**) seedling (**g**) development and (**h**) total. *ACT* = actin, *GAPDH* = glyceraldehyde-3-phosphate dehydrogenase, *EF1-alpha* = elongation factor 1 alpha, *SAMDC*= S-adenosylmethionine decarboxylase proenzyme, *UBC36* = ubiquitin-conjugating enzyme E2 36-like, *CYP* = peptidyl-prolyl cis-trans isomerase *CYP19-4*-like, *PTB* = polypyrimidine tract-binding protein homolog 1-like, *CAC* = intracellular transport protein, *PP2A* = serine/threonine protein phosphatase 2A, *ETIF3* = eukaryotic translation initiation factor3, *HSP* = protein microrchidia 2-like, *YLS* = thioredoxin-like protein.

**Figure 5 ijms-22-11017-f005:**
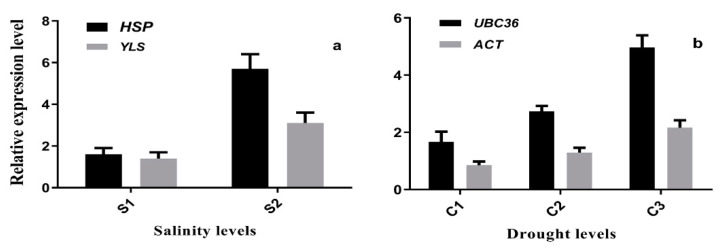
Relative expression level of target genes (*NHX1* and *DREB*) under salinity and drought stress, whereas salinity (S1 = 50 mM NaCl and S2 = 150 mM NaCl) (**a**) and drought (C1 = control, C2 = 75 g L^−1^ PEG-6000 and C3 = 100 g L^−1^ PEG-6000) (**b**) conditions.

**Table 1 ijms-22-11017-t001:** Primer sequence (forward and reverse 5′-3′), amplified characteristics, and 12 reference gene details for RT-qPCR.

Symbol	Gene Description	Gene Symbol in NCBI	Forward Primer (5′-3′)	Reverse Primer (5′-3′)	Amplicon Length (bp)	Correlation Coefficient	RT-qPCR Efficiency
*ACT*	Actin	LOC109759322	TTGCCTTGGATTATGAACA3^′^	GATGGCTGGAACAGAACTT3^′^	109	0.98	106%
*GAPDH*	Glyceraldehyde-3-phosphate dehydrogenase	LOC109783816	TACTTCACAGCCGATAGC	CCGATTGTTGGACGATACT	129	0.99	107.85%
*EF1-alpha*	Elongation factor 1 alpha	LOC109759403	ATTGGTGGCATTGGAACT	TCTCAACAGACTTAACCTCAG	115	0.99	111.60%
*SAMDC*	Phosphate2Aserine/threonine-protein phosphatase 2A	LOC109775250	TACTTCACAGCCGATAGC	CCGATTGTTGGACGATACT	114	0.98	117.12%
*UBC36*	Ubiquitin-conjugating enzyme E2 36-like	LOC109787298	GTATCACATAGACAGCATCATC	AGCATCCAGGACATTCAA	124	0.99	109.07%
*CYP*	Peptidyl-prolyl cis-trans isomerase CYP19-4-like	LOC109782168	GGTTAAGCCATCCTCCTT	CTCGTAGTTACAGTGGTGAT	116	0.98	111.44%
*PTB*	Polypyrimidine tract-binding protein homolog 1-like	LOC109755882	GACGATTCAAGCACGATTCT	TTGTTCCAGCCGATTGTTG	112	0.98	107.76%
*CAC*	Intracellular transport protein	LOC109744258	CACGCTGCTCACTGTTAC	AATCTGGCATCCTTCTACTTCA	92	0.98	112.24%
*PP2A*	Serine/threonine protein phosphatase 2A	LOC109772007	CTGTCACATAGGTAGAGTAGAT	CACGCAAGATGGAGTAAC	128	0.99	113.60%
*ETIF3*	Eukaryotic translation initiation factor 3	LOC109750994	AGCCTCTTCATCTTCTTCA	TTGGTCTGGATAGCATTGA	92	0.99	117.75%
*HSP*	Protein microrchidia 2-like	LOC109787360	CTGTATCTGAGGTTCTTGGT	CTACTCGGTGGTCAAACT	101	0.97	111.19%
*YLS*	Thioredoxin-like protein	LOC109773417	GTCAACGGTGATGTTCTTCT	ACAGTCTCCACAATGTCAAC	103	0.99	104.74%

**Table 2 ijms-22-11017-t002:** Reference genes expression stability ranked by BestKeeper under different abiotic stresses and at developmental stages in *Ae. tauschii*.

Gene	Control	Salinity	Drought	Heat	Cold	Seedling Stage	Development Stage	Total
*PP2*	0.5	0.29	0.54	2.76	1.87	0.42	2.09	2.55
*CAC*	0.46	0.06	0.49	1.85	1.49	0.6	1.97	1.43
*ETIF3*	0.52	0.45	0.65	3.68	1.63	0.43	1.55	3.37
*SAMDC*	0.24	0.22	0.71	0.18	1.16	0.17	1.54	2.67
*UBC36*	0.2	0.18	0.59	1.44	1.17	0.04	2.5	1.52
*EF1-alpha*	0.22	0.48	0.32	1.33	1.08	0.46	1.71	2.25
*HSP*	0.24	0.26	0.62	2.26	1.56	0.09	1.4	1.23
*GADPH*	0.42	1.16	0.46	1.78	0.73	0.13	1.1	1.53
*PTB*	0.39	0.51	1.04	1.71	1.32	0.27	2.1	1.51
*ACT*	0.52	0.38	2.27	2	1.17	0.97	1	1.54
*YLS*	0.64	2.21	1.03	0.58	1.04	0.45	1.85	1.75
*CYP*	0.9	0.17	0.38	1.66	1.27	0.97	0.81	2.24

*ACT* = actin, *GAPDH* = glyceraldehyde-3-phosphate dehydrogenase, *EF1-alpha* = elongation fac-tor 1 alpha, *SAMDC*= S-adenosylmethionine decarboxylase proenzyme, *UBC36* = ubiquitin-conjugating enzyme E2 36-like, *CYP* = peptidyl-prolyl cis-trans isomerase CYP19-4-like, *PTB* = polypyrimidine tract-binding protein homolog 1-like, *CAC* = intracellular transport protein, *PP2A* = serine/threonine protein phosphatase 2A, *ETIF3* = eukaryotic translation initiation fac-tor3, HSP = protein microrchidia 2-like, *YLS* = thioredoxin-like protein.

**Table 3 ijms-22-11017-t003:** Reference gene expression stability ranked by RefFinder under different abiotic stresses and developmental stages in *Ae. tauschii*.

Rank	Control	Salinity	Drought	Heat	Cold	Seedling Stage	Development Stage	Total
1	*UBC36* 1.78	*HSP* 1.78	*UBC36* 1.57	*UBC36* 1.78	*YLS* 1.78	*PP2A* 1.57	*ETIF3* 2.06	*CAC* 1.68
2	*SAMDC* 2.38	*UBC36* 2.28	*SAMDC* 3.22	*SAMDC* 2.38	S*AMDC* 2.83	*ETIF3* 2.3	*PTB* 2.58	*PTB* 1.73
3	*EF1-alpha* 3.46	*PP2A* 2.45	*EF1-alpha* 3.36	*EF1-alpha* 3.46	*ACT* 3.41	*YLS* 4.12	*PP2A* 3.98	*HSP* 2.24
4	*HSP* 3.98	*CAC* 3.50	*HSP* 3.98	*HSP* 3.98	*UBC36* 3.66	*EF1-alpha* 4.56	*EF1-alpha* 3.98	*GADPH* 3.87
5	*PP2A* 4.28	*ACT* 4.60	*GADPH* 4.28	*PP2A* 4.28	*GADPH* 4.41	*PTB* 5	*CYP* 4.41	*UBC36* 4.43
6	*CAC* 4.41	*CYP* 4.74	*CYP* 4.36	*CAC* 4.41	*PTB* 4.60	*UBC36* 5.62	*SAMDC* 4.68	*YLS* 5.63
7	*ETIF* 5.89	*SAMDC* 6.29	*CAC* 4.53	*ETIF* 5.89	*EF1-alpha* 5.18	*SAMDC* 5.63	*GADPH* 6.4	*ACT* 6.74
8	*GADPH* 6.62	*PTB* 7.52	*ETIF3* 7.09	*GADPH* 6.62	*CAC* 5.63	*HSP* 6	*CAC* 6.92	*EF1-alpha* 8.24
9	*PTB* 7.3	*ETIF3* 8.74	*PP2A* 7.54	*PTB* 7.3	*CYP* 8.21	*GADPH* 6.05	*HSP* 6.9	*CYP* 8.97
10	*ACT* 9.74	*EF1-alpha* 9.74	*PTB* 10.24	*ACT* 9.74	*ETIF3* 10.24	*CAC* 7.84	*ACT* 7.18	*SAMDC* 9.97
11	*YLS* 11	*GADPH* 11	*YLS* 10.74	*YLS* 11	*PP2A* 11.24	*CYP* 11	*YLS* 8.94	*PP2A* 10.74
12	*CYP* 12	*YLS* 12	*ACT* 12	*ETIF3* 12	*HSP* 11.47	*ACT* 12	*UBC36* 12	*ETIF3* 12

## Data Availability

The data presented in this study are available in manuscript and Appendix A.

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
