# Peer review of "Selection and Validation of Reference Genes for RT-qPCR Analysis in Aegilops tauschii (Coss.) under Different Abiotic Stresses"

_ijms, 2021, doi:10.3390/ijms222011017_

Round 1
Reviewer 1 Report
It is a well written original research paper about the selection and validation of reference genes for quantitative real-time PCR analysis in Aegilops tauschii under different abiotic stresses. No specific comments.
Author Response
Reviewer Response
Many thanks for your positive response; we also rechecked and revised the manuscript very carefully. Hopes now it's acceptable for publication. Thanks
Regards
Xiangju Li, Professor, Ph.D.
Weed Science
Institute of Plant Protection (IPP)
Chinese Academy of Agricultural Sciences (CAAS)
No.2 West Yuanmingyuan Road
Haidian, Beijing, China
Reviewer 2 Report
The major concern regarding this manuscript lies in the presentation of the results, where the written part is missing. Please write the chapter on Results. Without this, the manuscript is not complete.
The picture in the supplementary materials lacks basic information, such as band length and sample names.
lines 284-285 How long did the temperature treatments take and what duration was used for the temperature extremes? Supposedly not 10 days, as the next sentence inducates.
line 321 'NHX1 and DRP1 expression under salinity and drought stress condition'. The 2 genes mentioned above were GOI. What physiological role do they stand for? What were the reasons for choosing those genes for studying reference gene stability?
Quality of the black and white figures is low.
- There are figures where the last column cannot be fully seen.
- there are ones, which are elongated, probably due to editing issues
Minor issues
- line 68 '...but the number is small for weeds. ' please provide a reference.
- line 215. The sentence is doubled in the text. please remove. 'Expression stability evaluated by the geNorm NormFinder and BestKeeper was not completely constant, and this divergence was also reported in previous studies.'
Author Response
"Please see the attachment."

Reviewer 3 Report
Dear Authors,
The subject of the manuscript is interesting. I know from my own experience how important is the stability of a reference gene in expression analysis and how difficult it is sometimes to identify such a gene. However, I have strong reservations about the performance or description of the experiment.
First, the material is described imprecisely. In your description of the material, you state that the genotypes were collected in five different provinces in China. You need to give the locations of these sites. At the time of collection, were seeds taken from only one or many plants? Were SSD lines or heterogeneous accessions used for the study? This type of analysis should be performed on genetically homogeneous material so that plants under control and stress conditions have the same genetic background.
Has the tested material been previously characterized for resistance to the stresses under study? If so, what were the results, or where can they be accessed. Analyzing the stability of gene expression in material about which little or nothing is known can significantly complicate the interpretation of the result. It is not known exactly what conditions the plants grew under before being stressed and whether those conditions were stable. It is also unknown if the experiment was conducted simultaneously for all stresses. It is not stated how long the plants were subjected to the stresses. It is also unknown for how many sites/lines/genotypes the stability of reference gene expression under each stress condition was studied. If the analysis was not conducted for all, what was the selection criterion, and which materials were tested under specific conditions. I also have a problem with the homogeneity of the developmental stages studied. The stages given in the description, i.e. 2-3 leaves and 4-5 leaves, are neither precise nor do they indicate uniformity of the material studied.
There is also a lack of information on whether, when designing the primers, it was checked whether they came from a single or multiple exons to exclude amplification from genomic DNA residues.
In addition, I have other comments on other parts of the manuscript.
Line 33: Triticum aestivum instead of Triticum Aestivum
Line 44-45: This sentence is incorrect. RT-qPCR does not quantify RNA, it only quantifies the level of specific transcripts.
The graphs in Figure 2 are of the wrong type. The data shown there are not continuous, so the points should not be connected by a line.
The formatting of gene names should be corrected to italics throughout the manuscript. The manuscript should also be improved in a linguistic sense.
Best regards,
M.
Author Response
"Please see the attachment."

Round 2
Reviewer 3 Report
now it's ok.
Author Response
We thank you for useful comments and suggestions, which have significantly improved the above manuscript (ijms-1360880). We hope that the revised manuscript will be acceptable for publication in IJMS.
Thanks
Regards
Xiangju Li, Professor, Ph.D.
Weed Science
Institute of Plant Protection (IPP)
Chinese Academy of Agricultural Sciences (CAAS)
No.2 West Yuanmingyuan Road
Haidian,Beijing, China